# Anti-Melanogenesis Effects of a Cyclic Peptide Derived from Flaxseed via Inhibition of CREB Pathway

**DOI:** 10.3390/ijms24010536

**Published:** 2022-12-28

**Authors:** Ji Hye Yoon, Won Young Jang, Sang Hee Park, Han Gyung Kim, Youn Young Shim, Martin J. T. Reaney, Jae Youl Cho

**Affiliations:** 1Department of Biocosmetics, Sungkyunkwan University, Suwon 16419, Republic of Korea; 2Department of Integrative Biotechnology, Biomedical Institute for Convergence at SKKU (BICS), Sungkyunkwan University, Suwon 16419, Republic of Korea; 3Department of Plant Sciences, University of Saskatchewan, Saskatoon, SK S7N 5A8, Canada

**Keywords:** cyclic peptide, linusorb, flaxseed, melanin, anti-melanogenesis, CREB pathway, MITF

## Abstract

Linosorbs (Los) are cyclic peptides from flaxseed oil composed of the LO mixture (LOMIX). The activity of LO has been reported as being anti-cancer and anti-inflammatory. However, the study of skin protection has still not proceeded. In particular, there are poorly understood mechanisms of melanogenesis to LO. Therefore, we investigated the anti-melanogenesis effects of LOMIX and LO, and its activity was examined in mouse melanoma cell lines. The treatment of LOMIX (50 and 100 μg/mL) and LO (6.25–50 μM) suppressed melanin secretion and synthesis, which were 3-fold increased, in a dose-dependent manner, up to 95%. In particular, [1–9-NαC]-linusorb B3 (LO1) and [1-9-NαC]-linusorb B2 (LO2) treatment (12.5 and 25 μM) highly suppressed the synthesis of melanin in B16F10 cell lines up to 90%, without toxicity. LOMIX and LOs decreased the 2- or 3-fold increased mRNA levels, including the microphthalmia-associated transcription factor (MITF), Tyrosinase, tyrosinase-related protein 1 (TYRP1), and tyrosinase-related protein 2 (TYRP2) at the highest concentration (25 μM). Moreover, the treatment of 25 μM LO1 and LO2 inhibited the expression of MITF and phosphorylation of upper regulatory proteins such as CREB and PKA. Taken together, these results suggested that LOMIX and its individual LO could inhibit melanin synthesis via downregulating the CREB-dependent signaling pathways, and it could be used for novel therapeutic materials in hyperpigmentation.

## 1. Introduction

The skin is an essential external barrier because it defends the body against irritation from various factors such as pollution, temperature, and UV exposure. Skin can be largely divided into the epidermis, dermis, and subcutaneous fat layers [1,2,3]. Melanocytes exist in the basal layer between the epidermis and the dermis. The melanocytes produce their pigments, melanin, to protect the skin cells from external environments. The pigment synthesis starts with external stimuli such as UV array irritation, reactive oxygen species (ROS), and reactive nitrogen species in melanocytes [4,5]. In response, the alpha melanin stimulating hormone (α-MSH) binds to the melanocortin 1 receptor (MC-1R) [6] and initiates the Protein kinase A(PKA) and cAMP response element-binding protein (CREB) signaling pathway through the conversion of ATP to cAMP. In this pathway, CREB is considered a crucial transcription factor. It can be activated when PKA phosphorylates its Ser133 residue [7] and then binds to the cAMP response elements (CRE) of the promoter, which expresses the melanogenesis-related genes [8]. CREB also regulates the expression of genes involving cell survival, proliferation, and differentiation. Among the CREB signaling proteins, the microphthalmia-associated transcription factor (MITF) controls many proteins, including tyrosinase-related protein-1 (TYRP-1), tyrosinase-related protein-2 (TYRP-2), and tyrosinase as the transcription factor [9,10,11]. These genes induce the maturation of melanin, which is composed two ways, with eumelanin and pheomelanin [12,13,14]. The two matured melanin can be transported to keratinocytes in the form of melanosomes. In particular, Eumelanin has the ability to shield the UVR as a physical barrier. It acts as an absorbent filter that decreases the UV penetration to the epidermis [15]. This mechanism occurs to protect the skin from the UVB damage to the skin, but it can proceed uncontrollably in some diseases such as hyperpigmentation accompanied by the unnecessary secretion of melanin pigments [16]. In previous studies, the deregulation of MITF is known to cause skin diseases such as hyperpigmentation, melasma, and melanoma [17,18,19,20]. Thus, it is very important to maintain the proper expression level of MITF.

The effects of peptides derived from plants are in the limelight because of their superiorities to synthesized drugs and higher target specificity and selectivity [21]. Over the past few years, peptides surfaced as an alternative option compared to small molecule drugs. Anti-fungal, anti-viral, anti-bacterial, antiparasitic, antifeedant, anti-cancer, and anti-hypertensive-related immune system effects of peptides have been reported [22]. Durdam Das et al. clarified the plant-derived peptides from the PlantPepDB. According to this study, the cyclic peptide and orbitide represent 5.3% of the total plant-derived peptides (3848) [23]. Despite its excellent efficacy, the research was not well conducted, due to the scarcity of cyclic peptides. Therefore, we aimed to confirm countless possibilities of LO to be developed as the putative therapeutic agent for skin diseases.

LOMIX is a mixture of cyclic peptides derived from flaxseed. The cyclic peptide known as LO is composed of eight or nine amino acids, including phenylalanine, proline, leucine, isoleucine, valine, and tryptophan, in a head-to-tail structure. These components can be classified into [1–9-NαC]-linusorb B3 (LO1), [1-9-NαC]-linusorb B2 (LO2), [1–8-NαC],[1-(*R*_s_,*S*_s_)-MetO]-linusorb A2 (LO3), [1–8-NαC],[1-(*R*_s_,*S*_s_)-MetO]-linusorb B1 (LO4), and [1–8-NαC],[1,3-(*R*_s_,*S*_s_)-MetO]-linusorb A3 & [1–8-NαC],[1,3-(*R*_s_,*S*_s_)-MetO]-linusorb A1 (LO5 & 6) (Figure 1 and Table 1) [24]. Since this natural peptide has bioactive effects, it has been researched in various fields. For example, through cytotoxicity against a human breast cancer cell line and C6 glioma cell line, the cancer suppressor activity of LO was investigated [25,26]. Moreover, the mixture of LO, LOMIX, caused low nitric oxide (NO) production in RAW264.7 cells. Meanwhile, LOMIX acted as a treatment for various inflammation-mediated disease models, including gastritis, colitis, and hepatitis [27]. The production of NO was also one of the key regulators in melanogenesis. NO is one of the mediators related to melanogenesis and increases the TYRP-1 and tyrosinase levels [28,29]. In a previous report, the bioactive compound, quercetin 3-O-β-D-Glucuronide, which inhibits NO production, had anti-inflammatory effects and anti-melanogenesis effects [30]. Despite the superior efficacy of independent compounds and mixtures, the mechanism on the skin is still poorly understood. For these reasons, we investigated the anti-melanogenesis effects of LO at the cellular level.

## 2. Results

### 2.1. LOMIX Inhibited Melanin Secretion and Synthesis in B16F10 Melanoma Cells

Firstly, to identify the cytotoxicity of LOMIX on melanocytes, B16F10 melanoma cells, an MTT assay was performed with 0 to 400 μg/mL of LOMIX. As shown in Figure 2a, B16F10 melanoma cells did not demonstrate any cytotoxicity with 25–100 μg/mL of LOMIX. Second, we examined the activity of melanin synthesis in B16F10 melanoma cells with LOMIX treatments. The secreted melanin was investigated with the cultured medium of B16F10 melanoma cells. The secretion of melanin pigments was inhibited by the treatment of LOMIX 25–100 μg/mL in B16F10 melanoma cells stimulated with α-MSH for 48 h. To analyze the levels of intercellular melanin, the melanin contents were detected in the cell lysate of B16F10 melanoma cells. There were significant reductions of melanin synthesis in B16F10 melanoma cells treated with LOMIX over 50 μg/mL; LOMIX-treated cells demonstrated similar levels of melanin contents compared to non α-MSH-treated cells (Figure 2b,c). Since melanin level can be decreased by cell number, we again confirmed whether LOMIX can suppress cell cycle progress for 48 h. As Figure 2d,e show, LOMIX did not strongly alter cell numbers and each cell cycle stage. According to Kirsten Tief et al., blocking the tyrosinase enzyme activity causes low melanin pigment synthesis [31]. Therefore, to determine the regulatory effect of LOMIX on the activity of tyrosinase, the tyrosinase inhibition assay was performed. However, LOMIX did not demonstrate the inhibitory effect on the enzyme activity of tyrosinase (Figure 2f). As a result, it was suggested that LOMIX will target the upstream molecules of tyrosinase. Thus, we conducted the real-time PCR to verify the regulatory roles of LOMIX on the gene expression levels. The effects of LOMIX on the mRNA expressions of melanin pigmentation-related genes were examined in B16F10 melanoma cells. Interestingly, the mRNA expression levels of MITF, tyrosinase and TYRP1 were all downregulated in the LOMIX presence condition (Figure 2g–i). These results demonstrated that the components of LOMIX, LOs, could downregulate melanin synthesis by suppressing the factors involved in melanin synthesis at the gene levels. 

### 2.2. LOMIX Has a Variety of Cyclic Peptide as a Natural Complex

Based on the previous study from Shim et al. [24], the HPLC data on every single component of LOMIX, LOMIX was composed of six cyclic peptides, which are called LO1, LO2, LO3, LO4, and CLF&G. The structures of each cyclic peptide were shown in Figure 1. Moreover, the name of the chemical compound and the detailed ratio of LO were written in Table 1, respectively.

### 2.3. LO Showed No Toxicity to B16F10 Cells

Through the mass analysis of LOMIX, the contents of the single cyclic peptide LO could be identified. Thus, we tried to test a single component that specifically produces the anti-melanogenesis effects. The experimental concentration was decided by HPLC analysis, and the cell toxicity of each concentration was confirmed by MTT assay. The cell survival declined due to the treatment of 50 μM LO1 or LO2 (Figure 3a,b). However, except for these two components, LO3, LO4, and LO5 and 6 did not show toxicity at the concentration contained in LOMIX and at all concentrations (Figure 3c–e). Taken together, the cell viability of the B16F10 melanoma cells was not altered by the treatment of LO1 and LO2 with 25 μM, and LO3, LO4, LO5 and 6 with 50 μM.

### 2.4. LO Decreased Melanin Secretion and Melanin Synthesis in B16F10 Cells

We conducted the melanin secretion and contents assay in B16F10 melanoma cells to clarify the anti-melanogenesis mechanism of LO. The secretion of melanin, which was upregulated by α-MSH treatment, was dramatically downregulated by LO1 or LO2 presence in a dose-dependent manner (Figure 4a–d). The melanin synthesis was also decreased by two compounds in B16F10 cells. Whereas, the pigment secretion was only decreased with LO3 or LO4 (Figure 4e–h). Meanwhile, there are no significant effects on melanin secretion and contents about LO5 and 6 treatment (Figure 4i,j). Through the results, it was assumed that the decrease in melanin contents of LOMIX can be achieved through LO1 and LO2. Taken together, LO1 and LO2 significantly demonstrated the low synthesis of melanin in B16F10 melanoma cells, and LO3 and LO4 only decreased the melanin secretion to the media. In particular, we confirmed that LO1 and LO2 were the powerful anti-pigmentation reagents for B16F10 melanoma cells.

### 2.5. LO Downregulated mRNA Expression along the CREB Pathway

MITF is the transcription factor that regulates the genes involving TYRP-1, TYRP-2, and tyrosinase. Through the previous experimental results, it was suggested that LO1 and LO2 were the critical component of LOMIX for the pigmentation-reducing activity. To verify the effect of LOMIX in inhibiting melanin synthesis, mRNA expressions involved in melanin synthesis factors were confirmed in both LO1 and LO2. The real-time PCR was conducted in B16F10 melanoma cells with co-treatment of α-MSH and LO1 or LO2. The expression of MITF, TYRP1, and tyrosinase was effectively inhibited by the 24-h treatment of LO1 at mRNA levels in a dose-dependent manner (Figure 5a–d), and the same factors were also downregulated by LO2 administration (Figure 5e–h). In these experiments, LO1 and LO2 regulated the mRNA expression associated with the melanin synthesis.

### 2.6. Proteins Associated with Melanin Synthesis Decreased Following

Western blotting was conducted with B16F10 cell lysates that were treated with α-MSH for confirming the protein levels of upstream molecules to MITF. The protein expression of MITF demonstrated the same pattern with real-time PCR results. The levels of MITF were significantly decreased when LO1 was treated (Figure 6a). Following these results, the protein expressions of the CREB-dependent signaling pathway molecules were investigated. The phosphorylation of CREB was not only suppressed but also that of PKA by treatment of LO1 (Figure 6b). In the case of LO2 administration, decreased levels of MITF expression were demonstrated in B16F10 melanoma cells. (Figure 6c). We also checked the expression of upstream molecules such as CREB and PKA. The phosphorylation of CREB at Serine 133 was blocked by LO2 presence, not PKA (Figure 6d). In summary, LO1 and LO2 targeted the CREB pathway for further melanogenesis inhibition.

## 3. Discussion

Flax (*Linum usitatissimum* L.) has been cultivated in Canada, Russia, and China. It could be consumed in the form of fibers and seeds. It has been used in ayurvedic medicine and studied in various fields. Flaxseed has contributed to the health of human beings [32], and as a crude medicine for attenuating the symptom of cancers [25], alcoholic liver disease [33], intestinal injury [34], cardiovascular disease [35], and diabetes [36] because of various bioactive compounds, such as short-chain omega-3, lignan, polyunsaturated fatty acids, mucilage, and LO [37]. Among them, LO can be synthesized as the secondary metabolite from the post-translational modification in flaxseed. Each LO, which is, respectively, called LO1, LO2, LO3, LO4, and LO5 and 6, constructs eight to nine amino acids in a cyclic form (Figure 1). These cyclic peptides are commonly composite phenylalanine and proline. In particular, LO5 and 6 have [(R_s,_S_s_)MetO]-Leu-[(R_s,_S_s_)MetO] structures [24]. These various components have been researched for the treatment of disease. For example, its anti-cancer property was suggested in glioma cells [26] and reported therapeutic effects involving gastritis, colitis, and hepatitis in the mouse model [27]. Nitric oxide can affect to inflammation process by expanding the blood vessel. As a result, it makes other immune cells facilitate the infiltration to the inflammation sites [38,39]. Moreover, it relates to melanocyte biology, which synthesizes melanin pigments. NO acts as a melanogenesis stimulator and increases tyrosinase and TYRP-1 levels in UV-irradiation [28,29]. According to previous studies, we hypothesize that the NO scavenging activity of LO will be related to inhibiting tyrosinase activity. From these great investments and various effects, we investigated that LO could have the potential to improve skin conditions.

In our skin, there are numerous defenses against skin stimuli. Skin pigmentation is one of the defense mechanisms to prevent cell damage [40,41,42,43,44]. By transporting the melanin to keratinocytes, it can protect the inner skin cells from photodamage. However, unnecessary melanin production could be caused by pigmentary disorders such as hyperpigmentation, melasma, and solar lentigines. It has been reported that these diseases are affected by chronic or acute UV array exposure [45]. Therefore, we checked the melanin contents and secretion levels when LOMIX or LO and α-MSH were treated in B16F10 melanoma cells. The melanin synthesis and secretion inhibition activity of LOMIX were effective in all concentrations (Figure 2b,c) without altering cell numbers (Figure 2d,e). Moreover, LO-treated B16F10 melanoma cells demonstrated low melanin synthesis and transport in α-MSH-treated conditions. Especially, the treatment of LO1 or LO2 significantly decreased the pigmentation and secretion in B16F10 melanoma cells. (Figure 4a–d).

From the external stimuli, MC-1R, which recognizes α-MSH, activates the CREB signaling pathway. The first step is the conversion of ATP to cAMP. Through this process, the upstream molecules, including PKA and CREB, are phosphorylated and transcribe the downstream molecule, MITF [46]. The pivotal transcription factor in the CREB pathway, MITF, activates various genes related to melanin synthesis. When MITF binds their transcription site in nuclei, diverse proteins are coded, such as TYRP-1, TYRP-2, and tyrosinase [47,48,49,50]. These genes are in the cellular membrane to oxidase the melanin in four stages [40,44,51,52]. This mechanism is very important and natural, but the accelerated synthesis of melanin causes severe skin disorders such as hyperpigmentation or melanoma. Because of this importance, we searched for the anti-melanogenesis potential of LOMIX or LO. Ultimately, the administration of LOMIX downregulated the mRNA levels of MITF (Figure 2g), and the downstream molecules such as TYRP-1 and Tyrosinase were suppressed by LOMIX treatments (Figure 2h,i). In addition, MITF, TYRP-1, TYRP-2, and tyrosinase were also suppressed by LO1 or LO2 (Figure 5). The oxidation of melanin could be separated in two ways to form eumelanin and pheomelanin [53]. Mostly, Eumelanin is the black pigment that can be made in melanocytes. TYRP1 and TYRP2, which we mentioned earlier, are factors involved in eumelanin maturity [50,54]. From these results, we suggested that these compounds will be related to inhibiting the eumelanin mature process.

Since MITF is pivotal in melanogenesis, its heterozygous mutations caused pigmentary diseases such as Waardenburg syndrome IIA7 to demonstrate severe symptoms of deafness or hyperpigmentation [45]. It also regulated cellular response as the proliferation and survival of melanocytes, and the deregulation of MITF could be linked to pigmentary-tumor genesis [18,55,56,57]. The degradation of MITF was demonstrated when we treated LO1 or LO2 with α-MSH (Figure 6a,c). MITF can be regulated by the upstream molecules, CREB, which is phosphorylated by PKA activation [7]. The phosphorylated CREB expression was more decreased by the treatment of LO1 or LO2 than in other groups; the PKA activation was inhibited only by the LO1 treatment (Figure 6b,d). Therefore, our data suggested that LOMIX or individual LO could be applied to some diseases associated with MITF-deregulation as therapeutic agents through further studies.

How these individual LO can have different pharmacological activity should be further explored. For this, chemical polarity could be one of major driving forces to interact with the target molecule(s). In terms of this, LO1 and LO2 with lower polarity assumed by amino acid sequences such as LO3, LO4, LO5, and LO6 seem to have more accessibility to the target molecule. Moreover, the three-dimensional structural property of these molecules could be another critical factor. Thus, the chirality of methionine sulfoxide as side chains can affect its structural variance compared to the compound with an oxidized methionine (Figure 1). Therefore, these compounds (LO3–6) will have multiple and variable structures. It is supposed that some of these structures are inactive forms, implying that the active form of these compounds might be relatively lower than those of LO1/2. Since these compounds can be synthesized by the assemble of amino acids, additionally prepared artificial compounds can have a full understanding of its structural importance. Using the related genes, we are currently trying to prepare engineered LOs and will test again for their biological activities.

## 4. Materials and Methods

### 4.1. Materials

LOMIX and individual LO components (LO1, LO2, LO3, LO4, and CLF&G) were provided by Prairie Tide Diversified Inc. (Saskatoon, SK, Canada). B16F10 (murine melanoma cell line) cells were bought from the American Type Culture Collection (Rockville, MD, USA). Phenol red-free Dulbecco’s Modified Eagle’s medium (DMEM), normal DMEM, and streptomycin–penicillin solution were purchased from Hyclone (Logan, UT, USA). Fetal bovine serum (FBS) and trypsin-EDTA (0.25%) were obtained from Gibco (Grand Island, NY, USA). Phosphate-buffered saline (PBS) was purchased from Samchun Pure Chemical Co. (Seoul, Korea). Dimethyl sulfoxide (DMSO), mushroom tyrosinase, 5-hydroxy-2-hydroxymethyl-4H-pyranone (kojic acid), α-melanocyte-stimulating hormone (α-MSH), 4-hydroxyphenylalanine glucopyranoside (arbutin), and 3-(4,5-dimethylthiazol-2-yl)-2,5-diphenyltetrazolium bromide (MTT) were purchased from Sigma-Aldrich (St. Louis, MO, USA). Molecular Research Center, Inc. (Cincinnati, MA, USA) was the source of TRI Reagent^®^ for RNA preparation. Macrogen (Seoul, Korea) synthesized primers used in real-time PCR. HRP-conjugated goat anti-rabbit secondary antibody and horse anti-mouse antibody and primary antibodies targeting MITF, p-PKA, PKA, p-CREB, CREB, and β-actin were purchased from Abcam (Cambridge, UK), Santa Cruz Biotechnology (Dallas, TX, USA), or Cell Signaling Technology (Beverly, MA, USA). Enhanced chemiluminescence reagents were procured from Ab Frontier (Seoul, Korea).

### 4.2. Cell Culture

Mouse-derived B16F10 melanoma cells were incubated in the DMEM medium supplemented with 10% (*v*/*v*) FBS and 1% streptomycin–penicillin solution at 37 °C and 5% CO_2_. Sub-culture was conducted by detaching the cells with trypsin-EDTA after reaching 90% confluence.

### 4.3. Cell Viability Assay

The effects of LOMIX and each LO compound on cell survival were determined using the MTT assay. B16F10 melanoma cells (100 μL of 1.5 × 10^5^ cells/mL) were seeded in a 96-well plate with 100 μL of the testing sample [LOMIX (0–200 μg/mL) or LO (0–50 μM)]. After 2 days, MTT solution (10 μL) was added to each well, and cells were incubated for an additional 3 h in the dark. The reaction was stopped by adding 100 μL of 10% SDS complemented with 0.01 M HCl. Moreover, 5000 cells/well of B16F10 melanoma cells were incubated in 96-well plates to check the cytotoxicity of LOMIX in the presence of α-MSH. The 10 μL of MTT solution was added to the cells incubated with the LOMIX (50 and 100 μg/mL) and α-MSH (100 nM) for 0, 24, and 48 h. Subsequently, the stopping solution (100 μL) (10% SDS solution with 0.01 M HCl) was added after 3 h incubation. The absorbance was measured at 570 nm using Synergy™ HTX (Biotek Instruments Inc., Winooski, VT, USA).

### 4.4. Extracellular Melanin Secretion and Intracellular Melanin Content

B16F10 melanoma cells (1.5 × 10^5^ cells/mL) were seeded in a 6-well plate and cultured overnight. Then, the culture media were replaced with phenol red-free media containing DMSO (vehicle) or α-MSH (melanogenesis inducer) for the identification of the color change caused by melanogenesis. The cells were treated with different concentrations of LOMIX (25, 50, and 100 μg/mL) or LO A, B (6.25, 12.5, and 25 μM), D, E, and F and G (12.5, 25, and 50 μM). Arbutin at 1 mM was used as a positive control. After 48 h of the incubation, the culture medium’s optical density was detected at 475 nm to measure the amount of extracellular melanin secretion. Meanwhile, the remaining cells were harvested with 1 mL cold PBS and lysed with a lysis buffer (50 mM Tris-HCl pH 7.5, 120 mM NaCl, 20 mM β-glycerophosphate pH 7.5, and 2% NP-40). After centrifuging at 12,000× *g* for 5 min., the cell pellets were dissolved in 10% DMSO in 1 M NaOH and incubated at 60 °C. The melanin contents were measured at 405 nm using a Synergy™ HTX microplate reader. All photos of melanin-secreted medium and intracellular melanin contents were collected.

### 4.5. Cell Cycle Analysis

B16F10 melanoma cells (1.5 × 10^5^ cells/mL) were plated in 6-cm plates and incubated overnight. LOMIX (50 and 100 μg/mL) was then treated to the cells with **α**-MSH for 48 h to confirm whether LOMIX can affect cell proliferation during the melanin synthesis process. Subsequently, 4 mM of HU (hydroxyurea) was treated for 24 h to synchronize the cells and discarded. After further incubating for 2 d, the cells were washed with PBS and collected using the trypsin-EDTA solution. The cells were fixed with absolute EtOH and diluted after 20 min incubation at −20 °C. The supernatant of centrifuged cells was discarded. Continuously, cell pellets were incubated with 200 μg/mL of RNase for 30 min. Finally, PI staining solution (5 mg/mL) was added under light-blocked conditions for 30 min. The cell cycle analysis was performed by using the cytoFLEX Flow cytometer (Beckman coulter, USA) and CytFLEX software.

### 4.6. Tyrosinase Activity Assay

For this assay, 333 units/mL of mushroom tyrosinase were incubated with different doses of LOMIX or kojic acid (300 μM) at 37 °C and 5% CO_2_ for 30 min. Then, L-DOPA 2 mM was added for 5 min for further reaction. The optical density was measured at 475 nm with a Synergy™ HTX microplate reader.

### 4.7. mRNA Expression Measurement with Quantitative Real-Time PCR

The mRNA expression of melanogenesis-related genes was confirmed with the primers listed in Table 2. B16F10 cells (3 × 10^5^ cells/mL) stimulated by 100 nM α-MSH were treated with LOMIX (0–100 μg/mL), LO A or B (0–25 μM), or arbutin 1 mM for 24 h. The cells were lysed with 300 μL of TRI Reagent^®^ to extract total RNA. Then, 1 μg of total RNA with Reverse Transcriptase M-MuLV (Thermo Fisher Scientific, Waltham, MA, USA) was utilized to synthesize cDNA according to the manufacturer’s instructions. Real-time PCR was conducted, as previously described [58].

### 4.8. Immunoblotting Analysis

To induce melanogenesis, 3 × 10^5^ B16F10 melanoma cells per mL were treated with 100 nM of α-MSH. Subsequently, 1 mM of arbutin or LO A or B (0–25 μM) was administered to the culture media for 24 h. The cells were then harvested with 1 mL of cold PBS and lysed with lysis buffer containing protease inhibitors (20 mM Tris-HCl pH 7.5, 2 mM EDTA, 20 mM NaF, 150 mM NaCl, 50 mM β-glycerol phosphate, and 2% NP-40 containing 2 μg/mL of leupeptin, pepstatin, and aprotinin). The whole cell lysates were centrifuged at 12,000× *g* for 5 min to purify the intracellular protein. SDS-polyacrylamide gel electrophoresis (Bio-Rad, Hercules, CA, USA) was conducted to separate the equivalent protein contents according to size. The separated proteins were transferred to polyvinylidene fluoride membranes and blocked with 3% (*w*/*v*) bovine serum albumin (BSA) solution for at least 30 min at ambient temperature. They then were incubated overnight with primary antibodies (1:2500 dilution) at 4 °C. After three 10-min washes with 0.1% (*w*/*v*) Tween-20, the membranes were probed with HRP-conjugated secondary antibodies (1:2500 dilution). The signals of each protein sample were detected with enhanced chemiluminescence reagents.

### 4.9. Statistical Analysis

Data are represented as mean ± standard deviation (SD) of at least three replicates of separate experiments. The Mann-Whitney U test was used to determine the statistical differences between the control and experimental groups. Statistically significant values were considered at *p* < 0.05 (#, * *p* < 0.05; ##, ** *p* < 0.01).

## 5. Conclusions

Through our investigations, the specific mechanism of anti-melanogenesis was revealed, as summarized in Figure 7. LOMIX and their components as LO1 and LO2, induced the downregulation of pigmentation in B16F10 melanoma cells without toxicity. Moreover, LO3 and LO4 spurred the blocking activity in melanin secretion of the LOMIX. At the mRNA level, LOMIX contributed to the decreased expression of genes involved in melanin synthesis compared to arbutin, the tyrosinase inhibitor. According to the results, LO1 and LO2 demonstrated consistent results in that both compounds downregulated the mRNA level of MITF, Tyrosinase, TYRP-1, and TYRP-2. LO1 and LO2, respectively, inhibited the phosphorylation of PKA or CREB. Taken together, these studies suggested that LOMIX and its cyclicpeptide (LO) have an anti-melanogenesis effect via inhibiting the CREB signaling pathway. Moreover, current results strongly implicate a possibility that LOMIX can be developed as an anti-pigmentation drug or remedy to treat hyperpigmentation disorders such as Addison’s disease, Cushing’s disease, Acanthosis nigricans, Coeliac disease, and Graves’ disease [59,60,61]. For this, individual LO such as LO1 or a combination with two LOs (e.g., LO1 and LO2) can be also applied as a single drug or mixture drug of LO. The potential of individual LO or these mixtures of LOs will be further tested with a hyperpigmenting disease model in the following project. Finally, CREB is a functionally important transcription factor related to disease onset. Since the disturbance of CREB function is known to develop and progress Huntington’s disease, Rubinstein-Taybi syndrome, and major depressive disorder [62,63,64], we will also try to expand the use of CLMIX or its individual LO for treatment of these diseases.

## Figures and Tables

**Figure 1 ijms-24-00536-f001:**
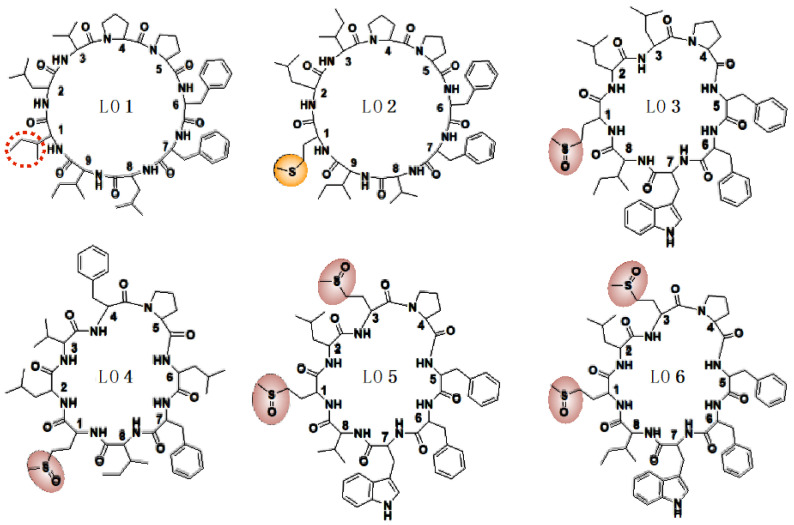
Chemical structure of each LO contained in LOMIX.

**Figure 2 ijms-24-00536-f002:**
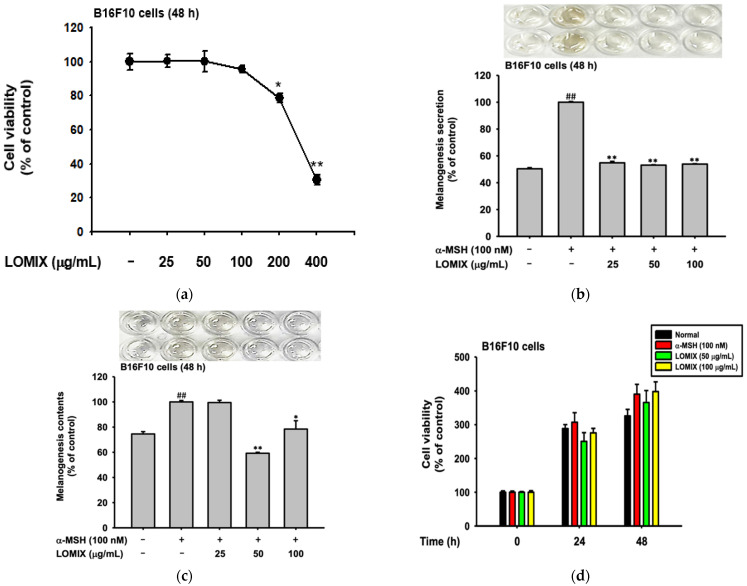
The activity of LOMIX in B16F10 cells. (**a**,**d**) B16F10 melanoma cells were treated at described concentrations of LOMIX for 0, 24, and 48 h, and cell viability was detected by MTT assay. (**b**,**c**) B16F10 cells pretreated with LOMIX were treated with α-MSH for 48 h, and melanin levels in cells (**b**) or media (**c**) were determined at 405 nm. (**e**) The cell cycle level controlled by LOMIX was determined by PI staining and Flow cytometry analysis with B16F10 melanoma cells treated with α-MSH and LOMIX for 48 h. (**f**) Tyrosinase inhibition assay was conducted using mushroom tyrosinase and L-DOPA. Kojic acid was used for positive controls in this experiment. (**g**–**i**) The mRNA levels of melanin production-related genes (MITF, Tyrosinase, and TYRP-1) were confirmed by a real-time PCR assay in B16F10 melanoma cells incubated with α-MSH and LOMIX for 24 h. (**a**–**i**) Values represent the mean ± standard deviation (SD) of three independent experiments. # *p* < 0.05 and ## *p* < 0.01 compared with normal via Mann-Whitney U test. * *p* < 0.05 and ** *p* < 0.01, compared with α-MSH-treated group via Mann-Whitney U test.

**Figure 3 ijms-24-00536-f003:**
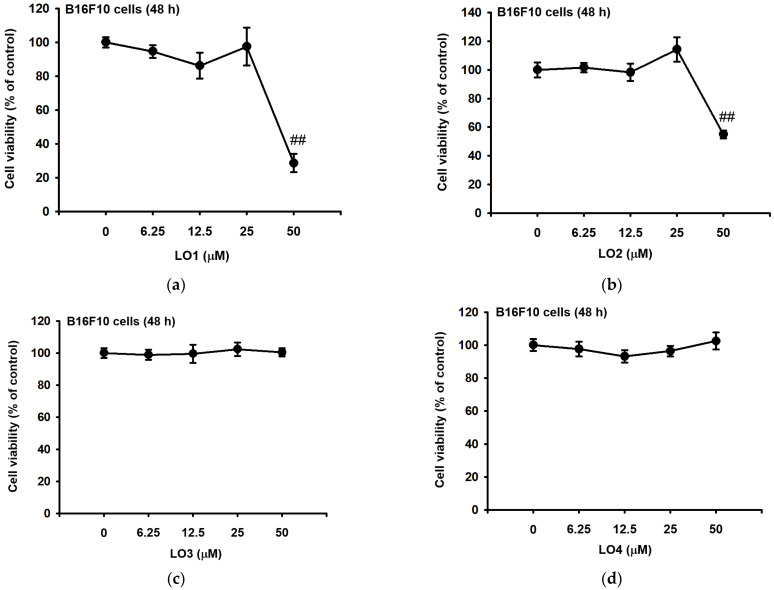
The cell viability was examined by conventional MTT assay. B16F10 cells melanoma were plated in 96well plates and treated with each LO. We confirmed the cytotoxicity of LO1 (**a**), LO2 (**b**), LO3 (**c**), LO4 (**d**), and LO5 and 6 (**e**) for 48 h. Values represent the mean ± standard deviation (SD) of three independent experiments. ## *p* < 0.01 compared with normal group via Mann-Whitney U test.

**Figure 4 ijms-24-00536-f004:**
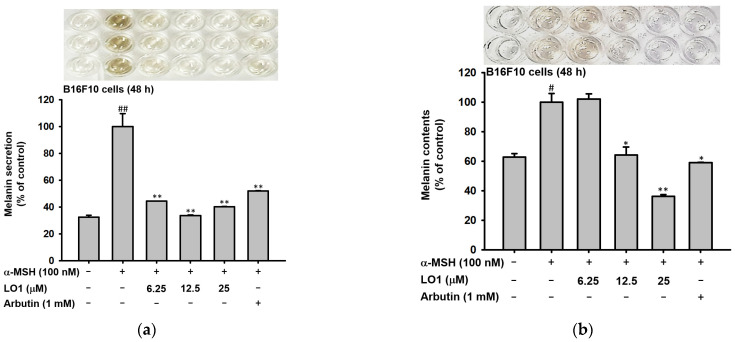
The melanin contents and secretion were detected in B16F10 melanoma cells. (**a**–**h**) The cells were treated with LO, which includes LO1, LO2, LO3, LO4, and LO5 and 6 with the presence of α-MSH at 48 h. Arbutin, which has been used as a tyrosinase inhibitor, was treated for positive control. After incubating, incubated media supernatants were detected for extracellular melanin at 405 nm (**a**,**c**,**e**,**g**,**i**). The cell pellets were dissolved by using the solution of 10% DMSO in 1N NaOH. The dissolvents were detected for intracellular melanin at 475 nm (**b**,**d**,**f**,**h**,**j**). Values represent the mean ± standard deviation (SD) of three independent experiments. # *p* < 0.05 and ## *p* < 0.01 compared with normal via Mann-Whitney U test. * *p* < 0.05 and ** *p* < 0.01, compared with α-MSH-treated group via Mann-Whitney U test.

**Figure 5 ijms-24-00536-f005:**
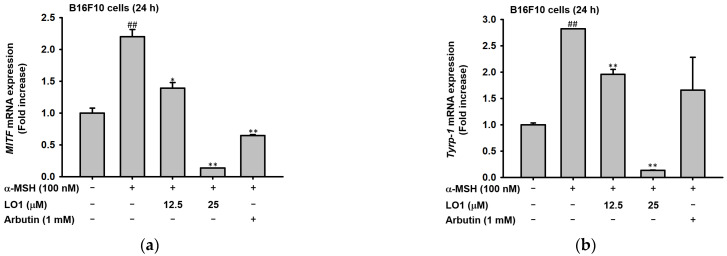
Real-time PCR was conducted to measure mRNA expression after α-MSH treatment accompanied by LO1 and LO2 treatment for 24 h in B16F10 melanoma cells. Expression of MITF mRNA level in LO1-treated condition in B16F10 melanoma cells. (**a**) The mRNA level of TYRP-1 with or without α-MSH. (**b**) The TYRP-2 expression at the transcription level when treated with LO1. (**c**) Tyrosinase expression was detected in a Real-time PCR assay in treating LO1. (**d**) MITF mRNA level when LO2 treated in B16F10 melanoma cells. (**e**) The expression level of TYRP-1 at the mRNA level. (**f**) The mRNA expression of TYRP-2 presence of α-MSH in B16F10 melanoma cells. (**g**) The expression of Tyrosinase in B16F10 melanoma cells cotreated with α-MSH and LO2. (**h**) Values represent the mean ± standard deviation (SD) of three independent experiments. # *p* < 0.05 and ## *p* < 0.01 compared with normal via Mann-Whitney U test. * *p* < 0.05 and ** *p* < 0.01, compared with α-MSH-treated group via Mann-Whitney U test.

**Figure 6 ijms-24-00536-f006:**
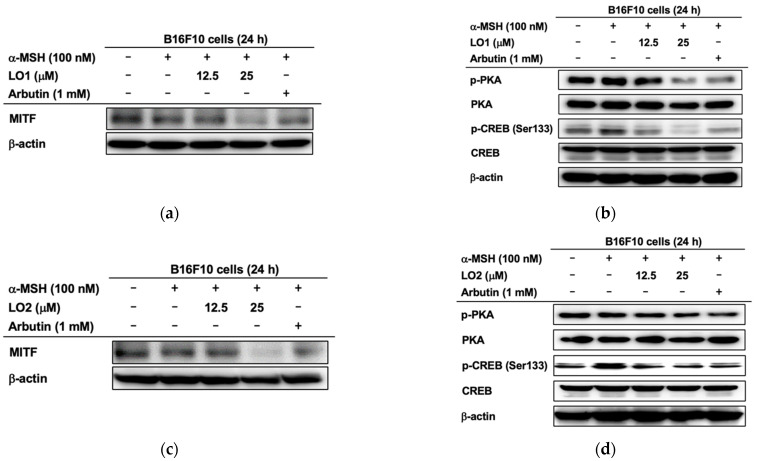
Effect of LO1 and LO2 on the level of melanogenesis-regulatory proteins in α-MSH-treated B16F10 melanoma cells. (**a**–**d**) The total or phosphorylated protein levels of melanogenesis-regulatory proteins, such as MITF, PKA, and CRAB, were detected by immunoblotting analysis.

**Figure 7 ijms-24-00536-f007:**
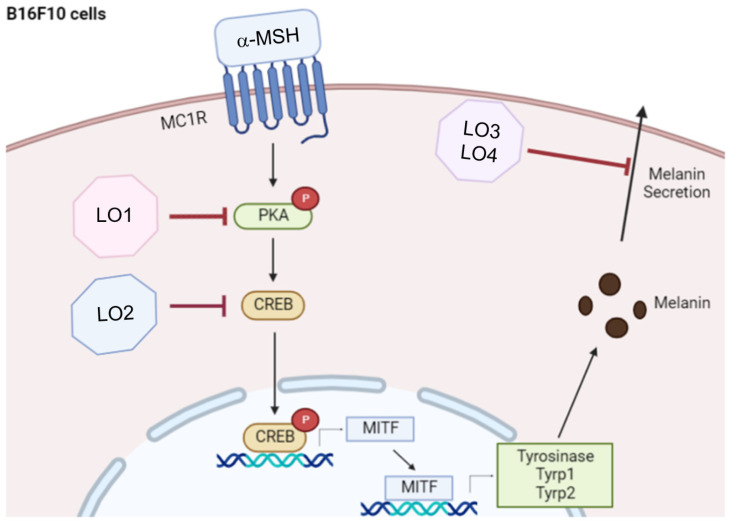
Summary of anti-melanogenic activity of LOs in α-MSH-treated B16F10 cells.

**Table 1 ijms-24-00536-t001:** The abbreviations of components in LOMIX and its details.

Code	LO Name	Literature Name	Molecular Weight (Da)	Quantity ^a^	Amount (μg/mL)
LO1	[1–9-NαC]-linusorb B3	CLA	1040.34	0.14 mg (23.0%)	46
LO2	[1–9-NαC]-linusorb B2	CLB	1074.37	0.16 mg (26.2%)	53
LO3	[1–8-NαC],[1-(*R*_s_,*S*_s_)-MetO]-linusorb A2	CLD	1064.34	0.05 mg (8.2%)	16
LO4	[1–8-NαC],[1-(*R*_s_,*S*_s_)-MetO]-linusorb B1	CLE	977.26	0.12 mg (19.7%)	39
LO5 & LO6	[1–8-NαC],[1,3-(*R*_s_,*S*_s_)-MetO]-linusorb A3	CLF	1084.35	0.03 mg (4.9%)	46
	[1–8-NαC],[1,3-(*R*_s_,*S*_s_)-MetO]-linusorb A1	CLG	1098.38	0.11 mg (18.0%)	

^a^ Quantity of LO1–LO6 in 200 μg/mL of LOMIX are analyzed with HPLC by Shim et al.

**Table 2 ijms-24-00536-t002:** Primer sequences for quantitative real-time PCR analysis.

Gene Name	Sequence (5′–3′)	
MITF	Forward	TCCGTTTCTTCTGCGCTCAT
	Reverse	CTGATGGACGATGCCCTCTC
TYRP-1	Forward	ATGGAACGGGAGGACAAACC
	Reverse	TCCTGACCTGGCCATTGAAC
TYRP-2	Forward	CAGTTTCCCCGAGTCTGCAT
	Reverse	GTCTAAGGCGCCCAAGAACT
Tyrosinase	Forward	GTCCACTCACAGGGATAGCAG
	Reverse	AGAGTCTCTGTTATGGCCGA
GAPDH	Forward	TGTGAACGGATTTGGCCGTA
	Reverse	ACTGTGCCGTTGAATTTGCC

## Data Availability

Not applicable.

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
