# Peer review of "Anti-Melanogenesis Effects of a Cyclic Peptide Derived from Flaxseed via Inhibition of CREB Pathway"

_ijms, 2022, doi:10.3390/ijms24010536_

Round 1

Reviewer 1 Report

A very well written and interesting study reports the anti-melanogenesis effects of linusorbs derived from flaxseed oil. It is certainly a vital topic as clinical trial and publications demonstrated that MITF-siRNA cream was used for the treatment for hyperpigmentation disorders, this study will provide a potential therapeutic intervention. A few questions remain to be addressed.

Point-by-point critique:

Figure 1. Does LOMIX has effect on the proliferation of melanocytes? Assay such as BrdU to exclude the possibility that the difference of melanin contents is not resulted by the cell numbers is vital.

Figure 2. Need to provide the discussion of the structure of the cyclic peptides and the function. Why CLA and B with a similar pattern of structure has a different effect on the melanocyte compared with CLE, F and G? Do these peptides have a similar bioavailability? Binding affinity? Do they have effects on the same targets on the pathways?

Figure 4. Author demonstrated that treatment of CLA or CLB significantly affects the function of B16F10 cells. However, it is not clear the effect of the treatment of LOMIX on cells. Activation and homeostasis markers such as C-kit needs to be determined. “The cell lysates were dissolved by using the solution of 10% DMSO in 1N NaOH.” It is cell pellets not lysates.

Last remark: what is missing is some further suggestions or recommendations by the authors on any future research. To prelude a bit more on further research, especially if it becomes a therapeutic agent, e.g. , Would even a combination of CLA and B be worth testing to explore if there is a potential synergy? If LOMIX has an influence on the CREB pathway, so what about other cell types? 

Overall, very supportive to get this study published.

Author Response

첨부 파일을 봐 주세요

Reviewer 2 Report

Dear authors,

I have reviewed your manuscript entitled "Anti-melanogenesis effects of a cyclic peptide derived from flaxseed via inhibition of CREB pathway"

your manuscript is interesting and if you pay attention to the following suggestions:

1. Clarify the abbreviated words such as LOMIX, CLA, and CLB, ...(Abstract section) if it is introduced for the first time in your manuscript, and list an abbreviation before the Introduction section.

2. B16-F10 is a cell line not just the cell, due to the cell line originating from primary cells, correct it.

3. I strongly suggest preparing your abstract in a quantitative manner, your abstract section is lack numbers, percentages, and level times, and ... you studied and have taken in your results.

4. Line 63: the authors claimed that: "Meanwhile, not only are they easy to synthesize, but...." if you are talking about the peptides derived from plants, hence you do not synthesize them, you should use extract or else instead synthesis, due to you do not synthesize them.

5. The quality of the figures is really low mainly the numbers in figures and I could not understand the figures without zooming in, improve them with save as images from the origin in MS word and resizing them.

6. I strongly suggest doing RSD, SD, or CV statistical analysis for all results that you have done for their means comparison. 

7. You cultured 1.5 x 105 cells/mL B16F10 in a 96-well cell culture plate, clarify how much media-containing cells you injected per well. 

Author Response

  1. Clarify the abbreviated words such as LOMIX, CLA, and CLB, ...(Abstract section) if it is introduced for the first time in your manuscript, and list an abbreviation before the Introduction section.
  • Thanks for your comments. CLA at al were concerned with the literature name which previous study nomenclature of their experiment. Therefore, we have revised the word and added the abbreviation according to the latest review paper. We revised name of these individual LO as below in the manuscript.

CLA-> LO1, CLB-> LO2, CLD-> LO3 CLE-> LO4, CLF&G->LO5&6

  1. B16-F10 is a cell line not just the cell, due to the cell line originating from primary cells, correct it.
  • Thanks for your comments. We have added “melanoma” to discriminate primary cell and cancer cell (see the manuscript).
  1. I strongly suggest preparing your abstract in a quantitative manner, your abstract section is lack numbers, percentages, and level times, and ... you studied and have taken in your results.
  • Thanks for your comments. We have added some of numeric in the Abstract section (see L17-31).
  1. Line 63: the authors claimed that: "Meanwhile, not only are they easy to synthesize, but...." if you are talking about the peptides derived from plants, hence you do not synthesize them, you should use extract or else instead synthesis, due to you do not synthesize them.
  • Thanks for your comments. Since it is not clear, we have deleted this sentence.
  1. The quality of the figures is really low mainly the numbers in figures and I could not understand the figures without zooming in, improve them with save as images from the origin in MS word and resizing them.
  • Thanks for your comments. We changed all figures and pictures more an improved version by enlarging the size and quality (see all figures).
  1. I strongly suggest doing RSD, SD, or CV statistical analysis for all results that you have done for their means comparison. 
  • Thanks for your comments. According to this comment, we have included all of statistical analyses (see L157-159).
  1. You cultured 1.5 x 105cells/mL B16F10 in a 96-well cell culture plate, clarify how much media-containing cells you injected per well. 
  • Thanks for your comments. We have revised it in this manuscript (see L323-324).